# microRNA Biology on Brain Development and Neuroimaging Approach

**DOI:** 10.3390/brainsci12101366

**Published:** 2022-10-09

**Authors:** Keita Tsujimura, Tadashi Shiohama, Emi Takahashi

**Affiliations:** 1Department of Radiology, Harvard Medical School, Boston, MA 02115, USA; 2Athinoula A. Martinos Center for Biomedical Imaging, Massachusetts General Hospital, Charlestown, MA 02129, USA; 3Group of Brain Function and Development, Nagoya University Neuroscience Institute of the Graduate School of Science, Nagoya 4648602, Japan; 4Research Unit for Developmental Disorders, Institute for Advanced Research, Nagoya University, Nagoya 4648602, Japan; 5Department of Pediatrics, Chiba University Hospital, Chiba 2608677, Japan

**Keywords:** miRNAs, gene expression, brain development, migration, neuronal development, neurodevelopmental disorder, magnetic resonance imaging

## Abstract

Proper brain development requires the precise coordination and orchestration of various molecular and cellular processes and dysregulation of these processes can lead to neurological diseases. In the past decades, post-transcriptional regulation of gene expression has been shown to contribute to various aspects of brain development and function in the central nervous system. MicroRNAs (miRNAs), short non-coding RNAs, are emerging as crucial players in post-transcriptional gene regulation in a variety of tissues, such as the nervous system. In recent years, miRNAs have been implicated in multiple aspects of brain development, including neurogenesis, migration, axon and dendrite formation, and synaptogenesis. Moreover, altered expression and dysregulation of miRNAs have been linked to neurodevelopmental and psychiatric disorders. Magnetic resonance imaging (MRI) is a powerful imaging technology to obtain high-quality, detailed structural and functional information from the brains of human and animal models in a non-invasive manner. Because the spatial expression patterns of miRNAs in the brain, unlike those of DNA and RNA, remain largely unknown, a whole-brain imaging approach using MRI may be useful in revealing biological and pathological information about the brain affected by miRNAs. In this review, we highlight recent advancements in the research of miRNA-mediated modulation of neuronal processes that are important for brain development and their involvement in disease pathogenesis. Also, we overview each MRI technique, and its technological considerations, and discuss the applications of MRI techniques in miRNA research. This review aims to link miRNA biological study with MRI analytical technology and deepen our understanding of how miRNAs impact brain development and pathology of neurological diseases.

## 1. Introduction

Properly formed and maintained neuronal morphology is critical for normal brain function [1]. Neuronal size and shape establish functional neuronal circuitry and connectivity. Various proteins/factors have been implicated in contributing to neuronal morphogenesis, including transcription factors, cytoskeletal elements, and components of various signaling pathways, however recently microRNAs (miRNAs) have also been considered important factors involved in the regulation of neuronal morphogenesis.

miRNAs are a new type of small non-coding RNA consisting of 19–24 nucleotides. miRNAs bind to the 3′ untranslated region (3′ UTR) of target messenger RNA (mRNA) and act as post-transcriptional regulators of gene expression [2,3]. The miRNA gene is primarily transcribed by RNA polymerase II to become primary-miRNA (pri-miRNA). It is then processed by the RNase III enzyme Drosha in the nucleus to produce precursor-miRNA (pre-miRNA). Pre-miRNAs migrate to the cytoplasm in an exportin-5-dependent manner and are further processed by Dicer which interacts with a double-stranded RNA-binding domain (dsRBD) protein, TAR RNA-binding protein (TRBP) into mature microRNAs. Mature miRNAs are subsequently incorporated into an effector complex called RNA-induced silencing complex (RISC), a ribonucleoprotein complex composed of argonaute (Ago) proteins, and mature miRNAs ultimately negatively regulate gene expression by inhibiting translation or promoting degradation of target mRNA [4] (Figure 1).

miRNAs have been reported to control various biological processes including apoptosis, growth, differentiation, and cell proliferation [5,6,7,8]. Several lines of evidence have revealed the crucial roles of miRNAs in the modulation of differentiation of neural cells and brain development [9,10,11].

Evaluating the brain morphology of patients and model mice is a pivotal step for judging the impact of impaired miRNA proportions on brain morphology at the organism level. Brain morphometry using anatomical brain magnetic resonance imaging (MRI) is a powerful modality with high spatial resolution, which could be attempted both *in vivo* and *ex vivo* beyond species [12].

In this review, we overview recent findings in the understanding of the biological roles of miRNAs in typical neuronal development and neurodevelopmental diseases including psychiatric disorders, and overview brain MRI approaches as a tool for evaluating the impact of impaired miRNAs on the brain morphology on the individual level.

## 2. miRNA Biology in Brain Development and Diseases

Extensive research has revealed a pivotal role in biological processes and disease pathogenesis. For example, it has been shown that certain miRNAs regulate cell proliferation and are involved in tumorigenesis [13]. In particular, microRNAs have been reported to be involved in the regulation of specific brain development and neural functions such as axon formation [14,15,16,17,18] and synaptogenesis [19,20,21,22,23,24,25,26], and in the causes and pathophysiology of neurological diseases such as neurodevelopmental and psychiatric diseases including autism spectrum disorder (ASD) [27,28,29] and major depressive disorder (MDD) [30,31]. Here we summarize the role of miRNAs in biological processes in the central nervous system and in disease pathogenesis of developmental disorders (Table 1 and Table 2).

### 2.1. Neurogenesis

Neurons are produced from neural stem cells and neural progenitor cells through a biological process called neurogenesis [68]. Neurogenesis is followed by migration, differentiation, formation of axons and dendrites, and synaptogenesis, all of which are essential for normal brain development and function. Therefore, neurogenesis is one of the fundamental processes for brain formation [68]. In mammalian development, the cerebral cortex becomes populated by inhibitory interneurons and excitatory projection neurons [69]. These inhibitory interneurons and excitatory projection neurons are produced in subventricular zones (SVZ) and proliferative ventricular zones (VZ) of the cortex that are located at the walls of the lateral ventricles of the brain. Excitatory neurons are derived from multipolar basal intermediate progenitors (bIPs) that have delaminated from the apical and basal surface and reside in the SVZ or are generated from apical radial glia (aRG) in the dorsal VZ in mice [68,70,71]. In humans, an RGs produce ununiform populations of proliferative basal progenitors (BPs), such as bIPs, and a second population of RG which lose their apical anchoring and move their cell body into the outer SVZ (oSVZ). This basal radial glia (bRG) has been shown to be important for cortical expansion and gyration [72,73,74]. On the other hand, in the distant medial and caudal ganglionic eminences (GEs), inhibitory GABAergic interneurons are specified. Within the mouse GE, a VZ and IP containing RGs and an SVZ containing numerous subapical progenitor cells (SAPs) develop, with the SAPs and IPs undergoing 60–70% of the total mitosis seen in the GE, thus expanding the pre-migratory interneuron population. [75].

Over the last few decades, studies have revealed gene and signal pathways that regulate neurogenesis [68,76]. On the other hand, recent studies have shown that miRNAs act as novel regulators of neurogenesis. let-7b, a let-7 miRNA family member, is expressed in the mammalian brain and is enriched during neurogenesis. let-7b has been shown to control the proliferation and differentiation of cortical progenitors by targeting TLX, a nuclear receptor that enhances cell cycle progression in the developing brain, and cyclin D1, a cell cycle regulator [32]. Consequently, let-7b promotes neurogenesis. Another study has reported that let-7b also positively modulates neurogenesis by targeting the High mobility group AT-hook 2 (Hmga2) in the mammalian retina [33]. MiR-124 is one of the miRNAs that are specifically expressed in the brain and has been shown to promote neurogenesis by targeting five negative regulators of neuronal differentiation such as Small CTD Phosphatases 1 (SCP1), BAF complex 53 kDa subunit (BAF53a), Polypyrimidine tract binding protein 1 (PTBP1), SRY-box containing gene 9 (SOX9) and Ephrin-B1 [34,35,36,37,38]. A more recent study reports that miR-124 is regulated by Down syndrome critical region 1 (DSCR1)-mediated ten-eleven translocation 1 (TET1) splicing to control adult hippocampal neurogenesis [77]. It has been shown that TET1 is a demethylase and controls the gene expression involved in both adult and embryonic neurogenesis [78,79]. miR-137 was found to promote positively neuronal differentiation and repress the proliferation of NSCs/NPCs in the SVZ of adult mouse brains [80]. In another report, miR-137 was shown to form a regulatory loop with TLX and LSD1, a transcriptional co-repressor of nuclear receptor TLX, in the NSCs/NPCs from the ventricular zone of embryonic mouse brains [39]. TLX recruits LSD1 to the promoter region of the miR-137 to repress miR-137 expression. Then, in turn, miR-137 targets and suppresses the LSD1 expression through 3′ UTR of LSD1 mRNA. Subsequently, enhanced miR-137 expression inhibits NSC proliferation and promotes neuronal differentiation. Other research showed that the methyl CpG binding protein 2 (MeCP2) associates with SRY-box containing gene 2 (Sox2) and suppresses miR-137 expression by binding to the miR-137 promoter. Altered miR-137 targets the enhancer of zeste homolog 2 (Ezh2), a histone methyltransferase that regulates the maintenance of the bivalent chromatin states of stem cells. Consequently, miR-137 controls the fate of NSCs through epigenetic regulation [40,81]. It has been reported that miR-9 regulates corticogenesis through the regulation of the proliferation and differentiation of cortical progenitors. miR-9-2/3 double-mutant mouse, which shows a remarkable decrease of mature miR-9 and its complementary miR-9* exhibits various abnormalities in telencephalic structures. In the mutant embryos, enlarged ventricles with hypoplastic cortical upper layers and decreased numbers of interneurons and Cajal–Retzius cells were found in the cerebral cortex. However, the radial glia scaffold was normally constructed in the embryos. Mechanistically, miR-9 was shown to target multiple transcriptional factors such as FoxG1 and Meis2 [41]. Also, miR-9 was reported to suppress NSCs proliferation and enhance neural differentiation by targeting TLX [42]. miR-34a was shown to target synaptotagmin-1 and syntaxin-1A to control neuronal differentiation under control by a p53-family member transcription factor TAp73 [43]. Liu et al. reported that High levels of miR-184 enhance proliferation but inhibit differentiation of adult NSCs. In addition, they showed that methyl CpG binding protein 1 (MBD1) represses miR-184 expression and miR-184 in turn negatively regulates Numblike (Numbl), a regulator of brain development, by binding to the 3′-UTR of Numbl mRNA [44]. Bruno et al. linked the miRNA and Nonsense-mediated decay (NMD) in the regulation of neuronal differentiation [45]. They reported that brain-specific miR-128 inhibits NMD by targeting the RNA helicase UPF1 and the exon junction complex core component MLN51 to control neuronal differentiation. Recent studies have shown that miR-128 modulates the proliferation and neurogenesis of cortical neural progenitors by targeting pericentriolar material 1 (PCM1) and miR-153 promotes neurogenesis by regulating the Notch signaling pathway through targeting of Jagged1 and Hey2 [46,47]. Furthermore, we recently reported that MeCP2 and its downstream target miR-199a promote neuronal differentiation of NSCs/NPCs by regulating BMP-smad signaling through targeting *Smad1* [26,48].

### 2.2. Neuronal Migration

During development, the migration of neurons is one of the key steps for proper brain development and function. A highly regulated and coordinated series of molecular and cellular events are needed to construct the different laminae of the cortex in rodents and humans [82]. The human cerebral cortex is the most highly developed brain region and plays a pivotal role in integrating and processing information from the entire body resulting in higher behaviors such as motor and social behaviors [83].

In the early developmental stages, newborn deep-layer excitatory neurons produced from the radial glial cells migrate radially from the VZ towards the marginal zone (MZ) undergoing somal translocation to form the cortical plate (CP). In the late developmental stage, As the cortex thickens, neonatal neurons change to multipolar migration until they reach the intermediate zone (IZ). In this process, multiple processes can be dynamically expanded and contracted. Then, multipolar cells shift into a bipolar shape [84]. After this transition, neurons begin directed radial migration through the CP and IZ using RG fibers as a migratory scaffold [84]. Then, upon reaching the outermost periphery of the CP, it switches to the terminal translocation mode. In this mode, the somata rapidly migrate in a radial glia-independent manner, terminating their migration just below the MZ, where they cross paths with previously born neurons [82,84]. Accordingly, the six layers of the cortex form in an inside-out and birthdate-dependent manner [85]. On the other hand, Interneurons initially migrate long distances into the cortex in two tangential streams. They then migrate radially and integrate into the various layers of the cortex [86,87,88,89].

The precise construction of the cortical layers by the migration of neurons is strictly regulated by various intracellular and extracellular signals and dysregulation of neuronal migration is implicated in various diseases [82]. Among these signals, Reelin signaling has been well studied and shown that Reelin signaling and its regulatory factors play a key role in neuronal migration [84]. In addition to these factors, several miRNAs have also been reported to control the migration of neurons. In an earlier study, in addition to the other functions in the nervous system, miR-9 was shown to inhibit neuronal migration of human NPCs from embryonic stem cells of human [49]. The study also reported that miR-9 targets and represses stathmin which increases microtubule instability and whose expression in human NPCs correlates inversely with that of miR-9. Rago et al. reported that miRNAs belonging to the miR-379-410 cluster are expressed in cortical progenitors and neurons, and the miR-379-410 cluster modulates neuronal migration during later developmental stages by targeting N-cadherin [50]. miR-128 controls the process of radial migration of neurons during the development of cortical lamination through the targeting Börjeson-Forssmann-Lehmann Syndrome gene *Phf6* mRNA [51]. Han et al. found that miR-19 enhances the migration of adult-born neurons by repressing Rapgef2 [52]. A recent study showed that miR-129 negatively regulates neuronal migration by targeting *Fmr1* mRNA which are transcripts of Fragile X syndrome gene *Fmr1* [53].

### 2.3. Axon Formation

Axons transmit information to other neurons by chemical signals. Therefore, the proper development of axons is necessary for a functional neural circuit. It is becoming clear that the presence of mRNA and its post-transcriptional regulation are important molecular mechanisms for axon development. Recent research has also revealed that miRNAs exist in axons and contribute to axon development.

Recent studies have identified several neuron-specific and brain-enriched miRNAs. It is known that miR-124 expression is upregulated during brain development, which is involved in axon elongation. [90]. A study reported that miR-124 modulates the development of axons by repressing LIM/homeobox protein 2 (Lhx2) expression [17]. Another study showed that miR-124 regulates axon growth by targeting mRNA for RhoG, a low molecular weight GTPase [15].

The brain-enriched miR-9 has been also well-examined among these miRNAs. Recent research reported that miR-9 is expressed in post-mitotic neurons and is detected in the axons of primary cortical neurons. miR-9 overexpression repressed the length of the axon and endogenous miR-9 inhibition had the opposite effect, suggesting that miR-9 inhibits elongation of the axon. These effects have been reported to occur via the regulation of microtubule-binding protein 1b (Map1b), a protein important for microtubule stabilization, a target of miR-9 [14].

A report by Zhang et al. showed the presence of miR-17-92 cluster components in cultured neurons, particularly in distal axons. Elevated expression levels of this cluster facilitated the development of axons through the reduction of phosphatase and tensin homolog (PTEN) expression, whereas the miR-19a inhibiting, one component of the cluster, repressed axonal development [18].

A recent study has revealed that miR-132 is a positive modulator of axonal formation in dorsal root ganglia (DRG) in mice. It has also been reported that miR-132 regulates the Ras GTPase activator Rasa1 expression [16].

### 2.4. Dendrite Formation

Proper dendrite growth and branching are critical for neuronal circuit function, as dendrites are the sites of most synaptic contacts. Recent evidence has shown that miRNAs are involved in the control of dendritic growth. This chapter outlines recent progress in elucidating the molecular mechanisms mediated by miRNAs that control dendrite development.

In early studies, Smrt et al. reported that brain-enriched miRNAs, particularly miR-137, act an important role in regulating dendritic growth. miR-137 overexpression negatively regulated dendritic morphogenesis, while blocking the function of miR-137 had the opposite action. It has been reported that Mind bomb one (Mib1), which is a ubiquitin ligase known to be important in neurodevelopment, controls this miR-137 effect on dendrite formation [54]. miR-9 has been reported to be involved in axon development and in regulating dendrite growth. The blocking of miR-9 causes impaired dendrite growth *in vivo* via repression of the transcriptional repressor RE1 silencing transcription factor (REST) [57]. It has been demonstrated that miR-132 controls the dendritic maturation of newborn neurons in the adult hippocampus, by regulating the GTPase-activating protein p250GAP by genetic experiment [56]. Another research showed that miR-134 is a component of miR-379-410, a large cluster of brain-specific miRNAs that are actively involved in dendrite development. miR-134 is a component of miR-379-410, a large cluster of brain-specific miRNAs. This miR-134 induces the development of dendrites by repressing the expression of the translational repressor Pumilio2 mRNA, an RNA-binding protein known to control dendritic formation [55].

Our recent study reported that miR-214 controls dendrite formation [58]. Our experiments revealed that miR-214 expression positively controls the development of dendrites while inhibiting one of the mature forms of miR-214, miR-214-3p represses dendrite growth. It has been also shown that miR-214-3p targets the conserved 3′-UTR of *quaking* (*Qki*), a schizophrenia risk factor.

### 2.5. Synaptogenesis

Several studies have shown that miRNAs play crucial modulators of synaptic plasticity and morphomechanics. Thus, it has been thought that miRNAs are fundamental to higher brain functions such as learning and memory [91].

Many miRNAs have been shown to function as negative modulators of synaptogenesis. For example, miR-134 was reported as the first miRNA to regulate synaptogenesis. miR-134 expression reduced dendritic spine size via translational repression of *Lim-domain containing protein kinase 1* (*LimK1*), a regulator of actin polymerization in cultured neurons [24]. It has been also reported that miR-34a represses synaptic function by targeting mRNAs of the synaptic components syntaxin-1A and synaptotagmin-1 [19]. A brain-enriched miRNA, miR-138 is reported to be localized in the dendrite and inhibits dendritic spine size via the repression of acyl protein thioesterase 1 (APT1), an enzyme regulating the palmitoylation status of multiple proteins that is shown to act at the synapse [25].

It has been also shown the positive modulation of synaptogenesis by miRNAs. miR-125b and miR-132 were reported to be linked with fragile X mental retardation protein (FMRP). Increased miR-125b expression resulted in longer and thinner processes of cultured hippocampal neurons, while expression of miR-132 induced stubby and mushroom-shaped spines [20]. miR-125b was also reported to inhibit its target expression, NMDA receptor subunit NR2A, along with argonaute 1 and FMRP. Another research revealed that miR-132 suppresses the expression of the Rho GTPase-activating protein p250GAP, supporting an active role in synapse formation *in vitro* and *in vivo* [22]. It has been reported that mRNA encoding the methyl CpG-binding protein 2 (MeCP2), a modulator of neuronal morphogenesis and synapse development is targeted by miR-132 [21,23]. In addition, our studies have reported that MeCP2 facilitates the miR-199a processing as a microprocessor Drosha complex component and that miR-199a positively controls excitatory synaptic density and transmission via repressing expression of mTOR signal inhibitors in the cultured neural stem cells and neurons [26,48].

### 2.6. The Pathogenesis of Neurodevelopmental and Psychiatric Diseases

Over the last few years, much research has reported that altered post-transcriptional regulation induced by dysregulation of miRNA might contribute to abnormal neuronal plasticity and function in diseases [92]. Particularly, miRNAs also have been linked to the pathogenesis of neurodevelopmental and psychiatric disorders [93]. Here, we summarize miRNAs that have been reported to contribute to the pathophysiology of neurodevelopmental and psychiatric diseases.

*MECP2* mutations cause Rett syndrome (RTT), a devastating neurodevelopmental disease. Recently, our research has reported a role for miR-199a in the pathophysiology of RTT [26,48]. Our evidence has revealed that MeCP2 promotes miR-199a processing as a Drosha complex component at the post-transcriptional level and miR-199a elevates mTOR signaling activity, which is implicated in a variety of neurodevelopmental and psychiatric diseases [94], by repressing expression of mTOR signaling inhibitors including *Sirt1, Hif1α*, and *Pde4d.* Also, we have shown that the genetic deletion of miR-199a recapitulates many phenotypes of RTT model mice. It has also been revealed the dysregulation of miR-199a expression at a post-transcriptional level in the RTT brain. In addition, we recently reported that MeCP2/miR-199a axis regulates neural stem/precursor cell differentiation by inhibiting bone morphogenetic protein (BMP)-Smad signaling through targeting *Smad1*, a downstream transcription factor of BMP, and that differentiation defects caused by dysregulation of MeCP2/miR-199a/BMP signaling pathway contributes to pathological conditions.

Duplication of the *MECP2* gene also leads to a severe neurodevelopmental disorder called *MECP2* duplication syndrome (MDS). MDS predominantly affects males and is characterized by a broad range of symptoms, including severe intellectual disability, seizure, speech abnormalities, hypotonia, developmental delays, and recurrent respiratory infections [95]. A recent study reported that brain and cultured neural progenitor cells (NPCs) of Tg (*MECP2*) transgenic mice that enhanced MeCP2 expression increases NPC differentiation into neurons and MeCP2 promotes the processing of miR-197 in NPCs derived from Tg mice. This work also showed that miR-197 targets the mRNA of ADAM10 and repressed its expression [59].

Autism spectrum disorder (ASD) is a common complex neurodevelopmental disorder that is characterized by impaired social communication, restricted and repetitive behavior, and limited interests. Accumulating study has suggested that genetic and environmental factors contribute to the pathogenesis of ASD, and the involvement of miRNA in ASD pathology has also been shown. miRNAs have been well studied for use as a biomarker of ASD [27,29]. Expression profiling of miRNA in lymphoblasts derived from ASD patients revealed differentially expressed miRNAs, such as miR-23a and miR-106b, that target genes significantly involved in neurological functions and disorders. Recently, high-efficiency whole-exome sequencing of Australian families with ASD identified rare single nucleotide variants within mature miR-873-5p sequences. The research group also showed that miR-873 variants have a 20–30% inhibition effect on candidate autism risk genes *ARID1B*, *SHANK3*, and *NRXN2* [28].

Angelman syndrome (AS) is a rare genetic and neurodevelopmental disorder characterized by severe developmental delay, intellectual disability, seizure, microcephaly, speech impairment, and movement disorder. AS is caused by the loss of function of a ubiquitin ligase *UBE3A* gene located at chromosome 15q11-q13. The *UBE3A* expression is paternally imprinted in neurons and loss of function of maternally inherited *UBE3A* cause AS [96,97]. AS model mice deficient in maternal Ube3a show various behavioral features of AS, including motor abnormality and cognitive impairment [98]. A recent study reported that miR-708 is downregulated in the brain of AS mice and miR-708 targets endoplasmic reticulum resident protein neuronatin leading to a decrease in intracellular Ca^2+^ [60]. These results suggest that mmiR-708-neuronatin-mediated aberrant calcium signaling might be implicated in AS pathogenesis.

miRNAs have also been shown to be involved in the pathogenesis of schizophrenia, one of the major psychiatric disorders. A genome-wide association study (GWAS) in schizophrenia patients reported that rs1625579, which is detected within the putative miR-137 primary transcript, is linked with an increased risk of schizophrenia [61]. Other groups have also confirmed this association [62,63]. Furthermore, it has been revealed that miR-137 variation specifically influences the activity of the posterior right medial frontal gyrus during a cognitive task and functional connectivity of the front-amygdala and dorsolateral prefrontal-hippocampus in emotional tasks by functional MRI (fMRI) studies [64,65]. A report has also revealed that elevated miR-137 expression levels induce the repression of presynaptic target genes including *synaptotagmin-1* (*Syt1*) and *complexin-1* (*Cplx1*) *in vitro* and *in vivo*, resulting in changes in synaptic plasticity and abnormality of synaptic vesicle trafficking [66]. Topol et al. have shown the role of reduced levels of miR-9 in a subset of schizophrenia patient-derived neural progenitor cells. This study has demonstrated a strong correlation between miR-9 regulatory activity and its expression, and miR-9 manipulation affects neural migration [67].

Major depressive disorder (MDD) is another high-prevalence psychiatric disorder and miRNAs have been reported to be implicated in MDD pathogenesis. Recently, research combining peripheral miRNA levels with MRI analysis has begun to be reported. Qi et al. have found that the expression levels of miR-132 in the peripheral blood of unmedicated patients with MDD are higher than those in healthy control subjects, and miR-132 dysregulation in MDD is associated with multi-facets of brain function and structure in the front-limbic network [31]. He et al. have reported that miR-9 is upregulated in the peripheral blood of MDD patients and abnormal expression of miR-9 is involved in altered amygdala connectivity in MDD by using fMRI [30].

## 3. Neuroimaging Approaches Using MRI Technology

As mentioned above, miRNAs have been shown to contribute significantly to brain structure and function and are essential for normal brain development. Therefore, dysregulation of miRNAs would cause a variety of changes in the diseased brain. Here, we review *ex-* and *in vivo* MRI both of which can be useful for investigating the biological functions of miRNA, especially during brain development, starting from fetal ages. MRI is widely recognized as the most useful modality for *in vivo* investigation of brain structures of humans and animal models because of its ability to image large areas, relatively high spatial resolution, high reproducibility, and minimally invasive nature. Although histology is superior to MRI in terms of spatial resolution and directness, MRI has an advantage over histology in comprehensively analyzing the whole brain (e.g., regional volumes and cortical curvatures), and in identifying global fiber connections. Unlike DNA and RNA expression, spatial expression of miRNAs in anatomical regions of the brain is unknown. Therefore, comprehensive studies of the entire brain, such as brain MRI, are necessary. In addition, morphological MRI analysis of the prenatal brain will play an important role in assessing the neurodevelopmental process due to genetic abnormalities.

### 3.1. Pros of Ex Vivo MRI over In Vivo MRI

Structural and diffusion *ex vivo* MRI provides images with a high spatial resolution (e.g., 100–800 μm, depending on the size of the specimen and scanners used) and high signal-to-noise ratio (SNR) [99,100]. MRI signal strength decreases with the distance from the coils. Therefore, high SNR and spatial resolution can be more easily aimed by using MRI scanners with high field strengths and custom-made radiofrequency (RF) coils that can be closely placed to the specimen [101,102]. Recently, efforts have been made to image the whole *ex vivo* adult brain at 100–200 μm [103,104]. However, in general, another advantage of *ex vivo* MRI is that we can process the sample into smaller pieces to obtain better images. For example, the location of the brain stem and cerebellum *in vivo* makes it challenging to obtain optimal SNR and spatial resolution of the cerebellar cortex and white matter pathways. However, in *ex vivo* MRI studies, the brainstem [105] and cerebellum [106] could be dissected from the cerebrum and scanned with an MR coil closely placed on the surface of the cerebellum, enabling the highest possible SNR and spatial resolution for the selected brain region.

Furthermore, the scan time of *in vivo* MRI is limited by the subject’s ability to remain still, while *ex vivo* MRI scan times take many hours to increase SNR and are limited almost exclusively by the availability of MRI scanners and their cost of use. Therefore, *ex vivo* MRI can detect information in detail about tissue properties that cannot be seen on *in vivo* MRI. *In vivo*, MR images tend to have poor image quality due to artifacts from subject motion [107,108] and air-tissue boundary around the ear/nasal cavities [109] in addition to low spatial resolution. *Ex vivo* MRI is not easily affected by motion artifacts, and susceptibility artifacts can be reduced by proper and careful sample preparation, such as avoiding bubble formation [110]. Thus, careful evaluation of high-quality *ex vivo* MR images in combination with histology can provide a basis for detecting subtle cortical abnormalities with T1w and T2w MRI and abnormal fiber tract development with dMRI.

Finally, the spatiotemporal structural changes of the human fetal brain that can be assessed by *ex vivo* MRI are complex processes due to spatiotemporal changes in gene expression and epigenetic modifications during development [111,112]. Studying the relationship between gene expression patterns and dMRI tractography at each gestation age would contribute to a better understanding of the spatiotemporal changes in human brain development (for reviews, e.g., [113,114,115,116]). *Ex vivo* MRI combined with genetic and histological analysis would be the first step toward a comprehensive understanding of the possible effects of gene expression on regional morphology and fiber growth/elongation.

### 3.2. Advances in Ex Vivo Structural MRI

*Ex vivo* structure MRI could provide images at the same time point as the histological examination of the tissue, ensuring the deep investigation of the neuropathological characteristics of brain abnormalities [117,118]. Importantly, e.g., regional brain volumes measured by *in vivo* and *ex vivo* MRI present a linear relationship [117].

The typical approach of *ex vivo* MRI employs autopsy-extracted brains and formalin-fixed by immersion [119]. This approach (*ex vivo ex situ* MRI) has a disadvantage in manipulations of extraction from the skull and fixation that potentially induce major and inevitable adverse effects for MRI scans [120]. These adverse effects are attributed to deformation by extraction [121], a gradient of fixation from surface to deep white matter [122], and artifacts by air bubbles trapped in the subarachnoid spaces [110]. To overcome the disadvantage, some investigators propose an alternative approach that employs the brain remaining inside the head as *ex vivo* in situ MRI [120]. Another progress in *ex vivo* MRI is whole-brain, ultra-high resolution MRI with 100–170 um isotropic resolution [103,123,124], which promises to bridge the gap between histology and conventional imaging [123].

On the other hand, in *in vivo* fetal MRI, the majority of protocols use single-shot T2w MRI sequences [125] that are robust to fetal motion. However, single-shot T2w scans tend to be inferior to multi-shot turbo spin echo (TSE) scans [126], which are often performed after birth. Furthermore, the TSE protocol as used *in vivo* is not optimal for characterizing the transient zones found in the fetal brain *ex vivo*. This is especially true in the final stages of fetal development, most likely due to differences in the tissue characteristics of the *in vivo* and *ex vivo* (fixed) brain. One way to address this issue is to obtain T2 maps of postmortem samples and infer the optimal imaging parameters for TSE that maximize T2 contrast [127].

### 3.3. Advances in Ex Vivo Diffusion MRI

*Ex vivo* dMRI is useful for detecting fine diffusion properties in tissues. Tractography techniques can be used to identify pathways throughout the brain mantle in three dimensions. Conventional DTI tractography remains practical when studying the overall picture and quantitative assessment of fiber pathways such as the corpus callosum, thalamocortical pathways, and cerebellar nodules, but can obscure fiber trajectories at crossing fibers. High-angular resolution dMRI (HARDI) [128,129] theoretically provides excellent angular resolution of fiber pathways traversing long distances in the brain with optimal scan parameters (e.g., high b-values). Importantly, *in vivo* dMRI tends to identify only major bundles, while *ex*
*vivo* dMRI clearly identified u-fibers and fine gray matter fibers along with long association pathways even in patients with neurological disease (Figure 2) [130]. This advantage in *ex vivo* dMRI leads to attempting a comparison between dMRI-derived fibers and histological findings.

Spatiotemporal maturation time-courses of the vestibular, cingulate bundle, corpus callosum, thalamocortical pathways, inferior longitudinal bundle, inferior anterior-posterior bundle, caudal bundle, arcuate bundle, as well as the insula, cerebellum, and many other pathways have been reported during the human fetal period using HARDI [100,131,132,133,134,135]. *Ex vivo* dMRI yields an exceptional resolution of fiber pathways, and comparable results can be obtained using *in vivo* imaging (e.g., [136]). Specifically, radial coherence in the cortex, which is identified in the human fetal brain using *in vivo* and *ex vivo* MRI, is consistent with the presence of radial glial fibers [136]. The high degree of reproducibility of *ex vivo* and *in vivo* results [105,137] may be important information for the clinical interpretation of this technique.

### 3.4. Translation of Ex Vivo Results to In Vivo Application

*Ex vivo* fetal brains are essential for evaluating histology and MRI findings, which can be used to guide the detection of cortical and fiber tract abnormalities *in vivo*. If an abnormal dMRI tractography pattern is detected *in vivo* and the same fiber pattern is detected in *ex vivo* dMRI tractography, the *in vivo* pathway is more likely to be a “real” pathway rather than noise or artifacts from MRI scans and/or dMRI tractography algorithms. However, the correlation between histology and MRI findings is often difficult to interpret. The reasons for this difficulty include the following points; (1) systematic correlation and coreference between histology and MRI images exist only for the human adult brain [138], (2) the definition of transient fetal zones on MRI images relies on scattered open histology slices that do not cover the entire brain [139,140,141,142,143], and (3) even some useful atlases covering the entire brain are based on descriptions suitable for the developing rodent brain. Thus, to our knowledge, no study has systematically examined MRI-histological correlations of the human fetal and child brain across the entire telencephalon. However, only the correlation of *ex vivo* MRI and histology can provide sufficient information for a biologically correct interpretation of *in vivo* MRI findings. Although *in vivo* MRI has lower resolution and unpredictability due to fetal movement compared to *ex vivo* MRI, *in vivo* fetal brain structural and diffusion MRI are well investigated [144,145,146,147,148,149,150]. Therefore, new MR image acquisition and reconstruction techniques [151,152,153,154] would allow detailed high-resolution *ex vivo* MRI atlas of the developing fetal brain, which can provide valuable information for the interpretation of *in vivo* fetal MRI findings and identify areas that still require technical improvements in MRI acquisition, reconstruction, or analysis.

Advances in *in vivo* fetal MRI techniques and interpretation directly contribute to advances in fetal therapies such as fetal surgery [155,156] and gene therapy [157,158]. Quantitative MRI analysis is essential not only for early diagnosis and evaluation of treatment efficacy but also for the assessment of treatment side effects, especially in the context of neurological diseases. Because neuronal migration and cortical gyration during brain development are closely related to spatiotemporal patterns of gene expression [159,160], we need to be aware of the potential risk of brain morphological modulation due to unexpected changes in gene expression patterns, especially in gene therapy need to be aware of potential risks of brain morphology modulation due to unexpected changes in gene expression patterns. Detailed *ex vivo* fetal MRI observations should aid in the interpretation of atypical morphological findings in various *in vivo* fetal MRI studies, including these fetal treatments.

For the investigation of infantile brains, the severity of prematurity and severity of identifiable brain lesions are related to smaller brain volumes, gray and white matter volumes at term equivalent age, childhood, and adolescence [161,162,163]. In addition, atypical brain maturation after preterm birth indicates that the growth trajectories of transient fetal zones may be affected. Advances in neonatal care have increased the survival rate of preterm infants, and the prevention of cognitive deficits that these children exhibit later in life has been arising as a renewed clinical proposition, and MRI-detectable biomarkers are beginning to serve as a useful tool for identifying cognitive deficits that these children exhibit later in life [164,165].

## 4. Concluding Remarks

Accumulating finding has revealed miRNAs are involved in brain development and disease pathogenesis (Table 1 and Table 2). Although functional analysis of miRNAs in the nervous system is still in its early stages, it has been suggested that miRNAs act as important modulators of neuronal and brain development and the etiology of neurodevelopmental disorders. Given the fact that numerous miRNAs are enriched or specific to the brain [166] and that one miRNA can target many genes [167], miRNAs may be considered as a group of molecules that govern the complex processes of neuronal development and brain function. Future studies are necessary to identify further functional targets of each miRNA and their downstream gene networks. Numerous studies have shown that the expression levels of many miRNAs are altered in various neurodevelopmental disorders [168,169]. Currently, however, functional studies of miRNAs *in vivo* are still limited. Therefore, elucidating the roles of these miRNAs will help to further understand the mechanisms of brain development and the pathogenesis of neurodevelopmental disorders, and open new avenues for designing therapeutic strategies targeting miRNA-mediated pathways in the CNS. Advances in *ex vivo* MRI techniques will improve the accuracy and comprehensiveness of identifying brain morphological changes caused by miRNA alterations across species. In the near future, combined miRNAs and MRI studies using animal models will be realistic. In addition to these animal studies, future studies using state-of-the-art MRI technology will increasingly reveal the importance of miRNAs in human brain development and disease pathogenesis. Also, MRI-based volumetric measures and specific miRNA changes have both been proposed as prognostic biomarkers for epilepsy [170]. However, the interrelationship between them is not yet known. Therefore, it is important to conduct research that covers both.

## Figures and Tables

**Figure 1 brainsci-12-01366-f001:**
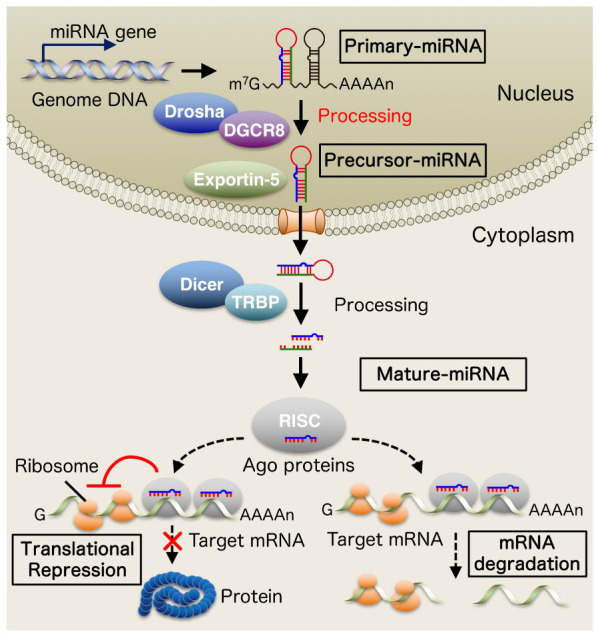
Biogenesis of microRNAs (miRNAs). The miRNA gene is primary transcribed by RNA polymerase II to become primary-miRNA (pri-miRNA). It is then processed by the RNase III enzyme Drosha in the nucleus to produce precursor-miRNA (pre-miRNA). Pre-miRNAs migrate to the cytoplasm in an exportin-5-dependent manner and are further processed by Dicer which interacts with a double-stranded RNA-binding domain (dsRBD) protein, TAR RNA-binding protein (TRBP) into mature microRNAs. Mature miRNAs are subsequently incorporated into an effector complex called RNA-induced silencing complex (RISC), a ribonucleoprotein complex composed of argonaute (Ago) proteins, and mature miRNAs ultimately negatively regulate gene expression by inhibiting translation or promoting degradation of target mRNA.

**Figure 2 brainsci-12-01366-f002:**
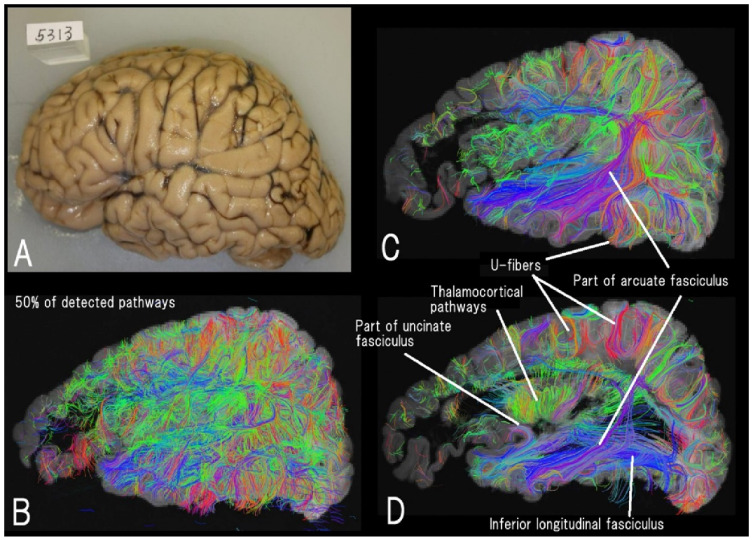
High-angular resolution diffusion MRI (HARDI) tractography of a left hemisphere (**A**) of a seven-year-old male with Alexander disease (**B**–**D**). Different levels of sagittal views of 50% of detected fiber pathways are shown. Thalamocortical, inferior longitudinal fasciculus, a part of the arcuate fasciculus and uncinate fasciculus, and many short-range cortico-cortical (u-fiber) pathways are clearly imaged (reproduced with permission to reuse [130]).

**Table 1 brainsci-12-01366-t001:** microRNAs (miRNAs) involved in brain development.

Function	miRNA	Effects	Targets	References
Neurogenesis	let-7b	Positive	TLX, cyclin D1	Zhao et al. [32]
			Hmga2	Xia and Ahmad, [33]
	miR-124	Positive	SCP1	Visvanathan et al. [34]
			BAF53a	Yoo et al. [35]
			PTBP1	Makeyev et al. [36]
			SOX9	Cheng et al. [37]
			Ephrin-B1	Arvanitis et al. [38]
	miR-137	Positive	TLX, LSD1	Sun et al. [39]
		Negative	Ezh2	Szulwach et al. [40]
	miR-9	Positive	FoxG1, Meis	Shibata et al. [41]
			TLX	Zhao et al. [42]
	miR-34	Positive	synaptotagmin-1, syntaxin-1A	Agostini et al. [43]
	miR-184	Negative	Numbl	Liu et al. [44]
	miR-128	Positive	UPF1	Bruno et al. [45]
	miR-153	Positive	Jagged1, Hey2	Qiao et al. [46]; Zhang et al. [47]
	miR-199a	Positive	*Smad1*	Nakashima et al. [48]
Migration	miR-9	Negative	stathmin	Delaloy et al. [49]
	miR-379-410 cluster	Positive	N-cadherin	Rago et al. [50]
	miR-128	Negative	PHF6	Franzoni et al. [51]
	miR-19	Positive	Rapgef2	Han et al. [52]
	miR-129	Negative	*Fmr1*	Wu et al. [53]
Axon formation	miR-9	Negative	Map1b	Dajas-Bailador et al. [14]
	miR-124	Positive	Lhx2	Sanuki et al. [17]
			RhoG	Franke et al. [15]
	miR-17-92 cluster	Positive	PTEN	Zhang et al. [18]
	miR-132	Positive	Rasa1	Hancock et al. [16]
Dendrite formation	miR-137	Negative	Mib1	Smrt et al. [54]
	miR-134	Positive	Pumilio2	Fiore et al. [55]
	miR-132	Positive	p250GAP	Magill et al. [56]
	miR-9	Positive	REST	Giusti et al. [57]
	miR-214	Positive	*Qki*	Irie et al. [58]
Synaptogenesis	miR-134	Negative	*Limk1*	Schratt et al. [24]
	miR-34a	Negative	synaptotagmin-1, syntaxin-1A	Agostini et al. [19]
	miR-138	Negative	APT1	Siegel et al. [25]
	miR-125b	Positive	NR2A	Edbauer et al. [20]
	miR-132	Positive	p250GAP	Impey et al. [22]
		Negative?	MeCP2	Klein et al. [23]; Hansen et al. [21]
	miR-199a	Positive	*Pde4d*, *Sirt1*, and *Hif1a*	Tsujimura et al. [26]

**Table 2 brainsci-12-01366-t002:** microRNAs (miRNAs) involved in the pathogenesis of neurodevelopmental and psychiatric disorders.

Disease	miRNA	Changes	Targets and Functions	Reference
Rett syndrome	miR-199a	Reduced expression in Rett syndrome brain and model	Targets *Pde4d*, *Sirt1*, and *Hif1a* to enhance mTOR signal activity	Tsujimura et al. [26]
		Reduced expression in the Rett syndrome model	Targets *Smad1* to repress BMP-Smad signaling	Nakashima et al. [46]
*MECP2* duplication syndrome (MDS)	miR-197	Enhanced expression in MDS model	Targets *ADAM10*	Wang et al. [59]
Autism spectrum disorder (ASD)	miR-23	Abnormal expression in ASD patient samples	Useful for biomarkers	Ghahramani Seno et al. [27]
	miR-106b	Abnormal expression in ASD patient samples	Useful for biomarkers	Sarachana et al. [29]
	miR-873	Abnormal expression in ASD patient samples	Targets *ARID1B*, *SHANK3* and *NRXN2*	Lu et al. [28]
Angelman syndrome (AS)	miR-708	Reduced expression in AS	Targets *Neuroatin*, leading to a reduction in intracellular Ca^2+^	Vatsa et al. [60]
Schizophrenia	miR-137	Enhanced expression in Schizophrenia patient samples	Targets *Cplx1* and *Sty1*	Schizophrenia Psychiatric Genome-Wide Association Study (GWAS) Consortium, [61]; Green et al. [62]; Potkin et al. [63]; Mothersill et al. [64]; Whalley et al. [65]; Siegert et al. [66]
	miR-9	Enhanced expression in Schizophrenia patient samples	Regulation of neural migration	Topol et al. [67]
Major depressive disorder (MDD)	miR-9	Enhanced expression in MDD patient samples	Useful for biomarkers	He et al. [30]
	miR-132	Enhanced expression in MDD patient samples	Useful for biomarkers	Qi et al. [31]

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
