# Peer review of "microRNA Biology on Brain Development and Neuroimaging Approach"

_brainsci, 2022, doi:10.3390/brainsci12101366_

Round 1

Reviewer 1 Report

The review provides an clear insight on the role of miRNA in the brain. The reviews is presented systematically with clear dissections in each section. The only disconnect in this review is the neuroimaging approach. There should be a better amalgamation of the two sections.

Line 356 MiRNAs - change case

Author Response

<Responses to the comments by Reviewer #1>

Comment 1-0

The review provides an clear insight on the role of miRNA in the brain. The reviews is presented systematically with clear dissections in each section. The only disconnect in this review is the neuroimaging approach. There should be a better amalgamation of the two sections.

Response 1-0

We appreciate the reviewer’s positive comments about whole of the manuscript. We also would like to thank the helpful reviewer’s comment which improve the quality of the statement. In accordance with the comment of Reviewer, we have added sentences in the Concluding remarks of our revised manuscript to emphasize that the two sections are related.

< Concluding remarks: Page 14, line 533-536>

Also, MRI-based volumetric measures and specific miRNA changes have both been proposed as prognostic biomarkers for Epilepsy. However, the interrelationship between them is not yet known. Therefore, it is important to conduct research that covers both.

Comment 1-1

Line 356 MiRNAs – change case

Response 1-1

In response to this reviewer’s comment, we have amended the capitalized case to lowercase “miRNAs” in the revised manuscript.

<2. miRNA biology in brain development and diseases; 2.6. The pathogenesis of neurodevelopmental and psychiatric diseases: Page 9, line 512>

Reviewer 2 Report

microRNA biology on brain development and neuroimaging approach (brainsci-1917526)
(Review)

The article by Tsujimura and colleagues provides a review on the role played by microRNAs (miRNAs) on brain development and function. Furthermore, considering the impact of miRNAs on aspects of brain development such as neurogenesis, migration, axon and dendrite formation, and synaptogenesis, the authors discuss the neuroimaging-based approaches to investigate the spatial expression patterns of miRNAs in the brain.

General Judgment Comments

The review is interesting for the readers and well structured. The title, abstract and keywords of the article frame correctly the manuscript in the existing literature. Although the content of the review is interesting, it is not clear the way in which the authors identified the articles from the existing literature. Some more work is needed on the format of Figures and Tables and on their captions to make them self-explanatory. Ultimately, several parts of the manuscripts include statements and facts that are not accompanied by the citation of sources of information.

I recommend the manuscript to undergo Minor Revision.

Major Issues

  • -  In the Introduction (mainly the first and second last paragraphs), please cite the original sources to justify your statements. References are needed to support the statements made by the authors.

  • -  The approach that authors used to identify the articles in the existing literature is not clear. Please clarify.

  • -  Lines 92-100, 180-188, 382-394, 411-421: please add citations to support your statements.

    Minor Issues

  • -  Figure 1: the figure’s title seems to be cropped, please fix it. The caption should be self-explanatory and abbreviation such as “miRNAs” should be used only after their extended version have been provided.

  • -  Please also fix the captions of Tables to make them self-explanatory.

  • -  Lines 80-81: “Extensive research has revealed a pivotal role in biological processes and disease pathogenesis”. A pivotal role played by what? Please clarify.

  • -  Line 82: “microRNAs have been reported to be involved in the regulation of specific brain development and neural functions”. Which neural functions? Provide examples and cite the related references in the main text too.

  • -  Line 82-83: “[...]and in the causes and pathophysiology of neurological diseases such as neurodevelopmental and psychiatric diseases”. Please provide examples on the disorders in which miRNAs is involved and cite the related references in the main text too.

  • -  Tables 1.1 and 1.2 are indexed in the text as Table 1 and Table 2. Please be consistent.

  • -  Table 1.1 doesn’t seem to be properly formatted. Some titles are in bold and come others are not (e.g., “target”, and “references”). Also, the title “function” and “miRNA” appear in the first row, while the other titles appear in the second row, replacing some content of the table.

  • -  Table 1.2: please be consistent in bolding the titles.

    Final comments

    I recommend the manuscript to undergo Minor Revision.

Author Response

<Responses to the comments by Reviewer #2>

Comment 2-0

The review is interesting for the readers and well structured. The title, abstract and keywords of the article frame correctly the manuscript in the existing literature. Although the content of the review is interesting, it is not clear the way in which the authors identified the articles from the existing literature. Some more work is needed on the format of Figures and Tables and on their captions to make them self-explanatory. Ultimately, several parts of the manuscripts include statements and facts that are not accompanied by the citation of the sources of information.

I recommend the manuscript to undergo Minor Revision.

Response 2-0

We appreciate the reviewer’s positive comments about the content of the review manuscript and we have amended the original manuscript in response to reviewer’s comments as described below.

Comment 2-1

In the introduction (mainly the first and second last paragraphs), please cite the original sources to justify your statements. References are needed to support the statements made by the authors.

Response 2-1

We would like to thank the helpful reviewer’s comment which improve the quality of the manuscript. In response to this reviewer’s comment, we have cited the original sources in the revised manuscript as follows:

<Introduction: Page 1, line 40>

< Introduction: Page 3, line 95>

Comment 2-2

The approach that authors used to identify the articles in the existing literature is not clear. Please clarify.

Response 2-2

For each topic, we have cited the most relevant papers as much as possible.If the reviewer has specific advice on how to improve our citation, we will gladly follow it.

Comment 2-3

Lines 92-100, 180-188, 382-394, 411-421: please add citations to support your statements.

Response 2-3

In response to this reviewer’s comment, we have added citations in revised manuscript.

Comment 2-4

Figure 1: the figure’s title seems to be cropped, please fix it. The caption should be self-explanatory and abbreviation such as “miRNAs” should be used only after their extended version have been provided.

Response 2-4

Figure 1 and legend has been modified in response to this reviewer's comments. The top title has been removed as it is no longer needed.

Comment 2-5

Please also fix the caption of Tables to make them self-explanatory.

Response 2-5

We have amended the caption of Tables.

Comment 2-6

Lines 80-81: “Extensive research has revealed a pivotal role in biological processes and disease pathogenesis”. A pivotal role played by what? Please clarify.

Response 2-6

In response to this reviewer’s comment, we have added a sentence and cited the related references.

For example, it has been shown that certain miRNAs regulate cell proliferation and are involved in tumorigenesis.

Comment 2-7

Line 82: “microRNAs have been reported to be involved in the regulation of specific brain development and neural functions”. Which neural functions? Provide examples and cite the related references in the main text too.

Response 2-7

We have added examples and cited the related references.

Comment 2-8

Line 82-83: “[...]and in the causes and pathophysiology of neurological diseases such as neurodevelopmental and psychiatric diseases”. Please provide examples on the disorders in which miRNAs is involved and cite the related references in the main text too.

Response 2-8

We have added examples and cited the related references.

Comment 2-9

Tables 1.1 and 1.2 are indexed in the text as Table 1 and Table 2. Please be consistent.

Response 2-9

In response to this reviewer’s comment, we have unified the format to Table 1 and Table 2.

Comment 2-10

Table 1.1 doesn’t seem to be properly formatted. Some titles are in bold and come others are not (e.g., “target”, and “references”). Also, the title “function” and “miRNA” appear in the first row, while the other titles appear in the secong row, replacing some content of the table.

Response 2-10

We have amended the titles of Tables.

Comment 2-11

Table 1.2: please be consistent in bolding the titles.

Response 2-11

We have amended the title of Tables.

Reviewer 3 Report

In this review article, the authors overviewed recent findings in the understanding of the biological roles of miRNAs in typical neuronal development and neurodevelopmental diseases including psychiatric disorders, and overviewed brain MRI approaches as a tool for evaluating the impact of impaired miRNAs on the brain morphology on the individual level.

Comments

This is an interesting study. The reviewer has some concerns as follows:

1. The format of entire manuscript needs to be revised, including the citation of references, list of references, and tables. As instruction to author: References must be numbered in order of appearance in the text (including citations in tables and legends) and listed individually at the end of the manuscript. Please also check the format of reference list.

2. Table 1 is in a chaotic state, and its format needs to be reorganized. Moreover, please unify the format in Table 1 and Table 2, and check the table titles if Table 1.1 and Table 1.2. are correct.

3. Please confirm if the top title (partially truncated) in the Figure 1 is required.

4. The descriptions in the Figure 1’s legend is confusing that “…RISC, a ribonucleoprotein complex composed of human immunodeficiency virus transactivating response RNA-binding protein (TRBP), argonaute 2 (Ago2), and Dicer,…”. It is really hard to see the relationship between this RISC complex and its upstream signaling molecules from the figure. The Figure 1 needs to be revised.

Author Response

<Responses to the comments by Reviewer #3>

Comment 3-0

This is an interesting study. The reviewer has some concerns as follows:

Response 3-0

We appreciate the reviewer’s positive comments and we have amended the original manuscript in response to reviewer’s comments as described below.

Comment 3-1

  1. The format of entire manuscript needs to be revised, including the citation of references, list of references, and tables. As instruction to author: References must be numbered in order of appearance in the text (including citations in tables and legends) and listed individually at the end of the manuscript. Please also check the format of reference list.

Response 3-1

In response to this reviewer’s comment, we have amended the format of reference list, tables and legends.

Comment3-2

  1. Table 1 is in a chaotic state, and its format needs to be reorganized. Moreover, please unify the format in Table 1 and Table 2, and check the table titles if Table 1.1 and Table 1.2. are correct.

Response 3-2

We have unified the format to Table 1 and Table 2.

Comment3-3

  1. Please confirm if the top title (partially truncated) in the Figure 1 is required.

Response 3-3

Figure 1 has been modified in response to this reviewer's comments. The top title has been removed as it is no longer needed.

Comment3-4

  1. The descriptions in the Figure 1’s legend is confusing that “...RISC, a ribonucleoprotein complex composed of human immunodeficiency virus transactivating response RNA-binding protein (TRBP), argonaute 2 (Ago2), and Dicer,...”. It is really hard to see the relationship between this RISC complex and its upstream signaling molecules from the figure. The Figure 1 needs to be revised.

Response 3-4

We apologize for the lack of clarity in the wording of the original manuscript. In accordance with the comment of Reviewer 3, we have amended Figure1 and legend.

<Figure 1 legend: Page2, line 58-64>

The miRNA gene is primary transcribed by RNA polymerase II to become primary-miRNA (pri-miRNA). It is then processed by the RNase III enzyme Drosha in the nucleus to produce precursor-miRNA (pre-miRNA). Pre-miRNAs migrate to the cytoplasm in an exportin-5-dependent manner and are further processed by Dicer which interacts with a double-stranded RNA-binding domain (dsRBD) protein, TAR RNA-binding protein (TRBP) into mature microRNAs. Mature miRNAs are subsequently incorporated into an effector complex called RNA-induced silencing complex (RISC), a ribonucleoprotein complex composed of argonaute (Ago) proteins, and mature miRNAs ultimately negatively regulate gene expression by inhibiting translation or promoting degradation of target mRNA.

Round 2

Reviewer 3 Report

This revised manuscript can be accepted. No further comments.